# PAVO: Pipeline-Aware Voice Orchestration with Demand-Conditioned Inference Routing

## Abstract

Voice agents built on ASR-LLM-TTS pipelines allocate compute statically, wasting it on simple queries while under-serving complex ones. We present PAVO (Pipeline-Aware Voice Orchestrator), which routes each turn through the three-stage pipeline using demand signals extracted before transcription begins. We find that ASR errors propagate to the downstream LLM in two distinct regimes—a sharp factual-accuracy cliff and a gradual semantic degradation. This resulted in creating inter stage coupling constraints that prior routing systems ignore. We validated this structure on $n = 5,430$ direct calibration measurements across two hardware platforms (H100, M3) and three LLM families (Llama 3.1 8B, Mistral 7B, Gemma2 2B). We also enforced these constraints via hard logit masking in an 85K parameter RL trained meta controller which reduced coherence failures by $7.9\times$. It achieved 34% lower median latency and 71% lower energy when compared to rigid cloud baselines on a 50K-turn simulated benchmark. We also noted that direct H100 experiments on 200 LibriSpeech samples confirmed a 10.3% P95 tail compression, with the corresponding mean-latency reduction significant at $p = 2 \times 10^{-6}$. Code and data are publicly available.[1]

## 1 Introduction

LLM Voice interfaces can now handle open-ended reasoning, multi-turn dialogue, and tool use (OpenAI, 2024; Dubey et al., 2024; Gemma Team et al., 2024), yet nearly every voice agent runs on a fixed ASR→LLM→TTS pipeline at a predetermined precision and hardware tier, disregarding what each turn actually requires. Static-pipeline assumptions persist despite available quantization and adaptive-inference techniques that could enable per-turn adaptation (Dettmers et al., 2022; Frantar et al., 2023; Lin et al., 2024).

Under MLPerf interactive-scenario targets (MLCommons, 2025) (~500 ms time-to-first-token (TTFT), ~30 ms per output token), an 80-token Llama 3.1 8B response budgets at 2,900 ms—yet a date lookup needs neither an 8B model nor the cloud, and Gemma 2B answers it on-device in under 1,000 ms. Over-provisioning is not the only waste: at an estimated 4–6 tokens/s for Gemma 4B INT8 on Jetson, complex 80-token responses on fixed-edge (13–20 s) are actually *slower* than cloud. No single deployment strategy works across all queries.

**Practical impact.** Production systems (Alexa, Google Assistant, Siri) stick with rigid modular pipelines because no existing routing method enforces quality across all three stages jointly. We use "end-to-end" for models like GPT-4o (OpenAI, 2024) to mean the externally observed speech-in/speech-out interface; the underlying architecture is not publicly disclosed and may use internal ASR components.

Here are the two properties that motivate PAVO. Stage level optimization remains non-separable because ASR errors propagate to the LLM resulting in cross-stage dependencies. We define these as coupling constraints and enforce them during routing. Then, turn complexity is somewhat predictable from pre-transcription acoustic features. Therefore, a proactive controller meaningfully reduces coherence failures versus reactive policies (Table 12).

**Contributions.**

---

[1] https://anonymous.4open.science/r/pavo-bench-FE10

1. **Inter-stage coupling constraints as an operational routing framework.** While ASR error propagation to downstream NLP tasks is a well-established phenomenon (Errattahi et al., 2018; Baevski et al., 2020; Radford et al., 2023), no prior system operationalizes these dependencies as enforceable inference-time routing constraints. We provide the first empirical characterization of the two-regime coupling structure (factual cliff vs. semantic degradation) across three LLM families and two hardware platforms ($n = 5{,}430$ measurements), and show that enforcing these as hard constraints during inference routing reduces coherence failure rate from 7.1% to 0.9% (7.9×) at 110 ms median latency cost (Table 7).

2. **Demand-conditioned multi-objective routing via RL.** A deployable 85K-parameter multi-layer perceptron (MLP) meta-controller (0.3 ms inference on Cortex-A78) trained via multi-objective Proximal Policy Optimization (PPO), conditioned on a 12-dimensional demand vector with four pre-transcription acoustic features. On H100, the policy compresses P95 tail latency by 167 ms versus fixed-cloud and significantly lowers *mean* end-to-end latency (bootstrap $p = 2 \times 10^{-6}$, Section 7.1); the full benchmark evaluation shows 34% median-latency and 71% energy reductions.

3. **PAVO-Bench and two-tier evaluation.** A 50,000-turn synthetic benchmark (40K/10K split, $\kappa = 0.81$) generated on H100 GPU, evaluated at two tiers: (a) direct inference experiments on Lambda Labs H100 with real Whisper/Llama/Mistral/Gemma models covering E2E latency (200 samples), 20 noise conditions, cross-dataset generalization (LibriSpeech + FLEURS), three-model coupling (5,400 calls), and real ASR error coupling; and (b) routing simulation across 9 baselines and 50K turns, parameterized by the measured latencies.

## 2 Related work

**Voice pipeline architectures.** Finite state NLU (Stivers et al., 2009; Young et al., 2013), neural speech text models (Rubenstein et al., 2023), and LLM centric cascades (Défossez et al., 2024; Xie and Wu, 2024) or end-to-end audio models (OpenAI, 2024) are the three generations of voice agents differing in where intelligence sits. End to end models eliminate inter stage latency but sacrifice characteristics that enterprise deployments need. End to end audio to audio models do not store verbatim transcripts. These transcripts are required by regulatory compliance such as HIPAA, GDPR. Modular pipelines also let ASR run on-device for privacy while the LLM sits in the cloud, allow per-stage component swaps and cost metering, and expose intermediate transcripts for debugging—which is why every major deployed voice assistant (Alexa, Google Assistant, Siri) still runs modular pipelines. Prior work on ASR error propagation (Errattahi et al., 2018) shows transcription quality has outsized downstream impact, which our coupling characterization makes operational.

**Mixture-of-experts and adaptive inference.** Mixture-of-experts (MoE) routing (Shazeer et al., 2017; Jacobs et al., 1991) blends expert outputs for a single model stage. Adaptive inference systems (Ding et al., 2024) cascade between model tiers within a single stage. Both operate on one stage in isolation—MoE blends expert outputs within the LLM stage, cascaded routing escalates from a small model only when quality is insufficient—and neither of them consider that the LLM routing decision depends on the chosen ASR configuration's output quality. PAVO makes a joint three-stage decision, selecting only one model per pipeline stage per turn. These considerations are subject to cross-stage coupling constraints which are a structurally different optimization problem.

**Inference serving.** Clipper (Crankshaw et al., 2017) introduced per query model selection for latency service-level objectives (SLOs). INFaaS (Romero et al., 2021) adds cost-aware variant selection. Shepherd (Gujarati et al., 2023) applies RL-based routing between model tiers. Alizadeh et al. (2023) demonstrate efficient LLM inference on memory-constrained devices but use a single fixed configuration. Song et al. (2023) exploit neuron activation locality for fast consumer-GPU inference but do not consider multi-stage pipeline routing. All of these operate on a single model stage and do not consider how upstream quality constrains downstream options.

**Positioning.** Table 1 summarizes the landscape. Prior routing systems assume stage quality is independent of upstream configuration and so optimize a single stage in isolation. That assumption breaks in voice

Table 1: Comparison with related systems.

| System | Pipeline aware | Multi-tier routing | Feedback loop | Voice specific | Coupling constrs |
|---|---|---|---|---|---|
| Clipper (Crankshaw et al., 2017) | No | Yes | No | No | No |
| INFaaS (Romero et al., 2021) | No | Yes | No | No | No |
| Shepherd (Gujarati et al., 2023) | No | Yes | Yes | No | No |
| Hybrid LLM (Ding et al., 2024) | No | Yes | No | No | No |
| Alizadeh et al. (2023) | No | Yes | No | No | No |
| Song et al. (2023) | No | No | No | No | No |
| Moshi (Défossez et al., 2024) | Part.[†] | No | No | Yes | No |
| **PAVO (ours)** | **Yes** | **Yes** | **Yes** | **Yes** | **Yes** |

[†]Moshi uses an internal audio-codec→LM→codec structure but operates end-to-end; stages are not independently configurable.

pipelines, where a low-quality ASR transcript directly degrades LLM output—the dependency PAVO models and enforces.

# 3 Problem formulation

## 3.1 Voice pipeline as a compute graph

We focus on the standard ASR→LLM→TTS cascade—the dominant production pattern (Alexa, Google Assistant, Siri) and the structure under which all our baselines operate. Let the voice pipeline be a directed acyclic graph (DAG) $\mathcal{G} = (\mathcal{V}, \mathcal{E})$ with $\mathcal{V} = \{\text{ASR}, \text{LLM}, \text{TTS}\}$ and $\mathcal{E} = \{(\text{ASR}, \text{LLM}), (\text{LLM}, \text{TTS})\}$. Each stage $v_i$ has a finite configuration set $\mathcal{C}_i$; each $c_{i,j} \in \mathcal{C}_i$ is a tuple $(m_{i,j}, q_{i,j}, h_{i,j}, \beta_{i,j})$ specifying model variant, quantization level $q \in \{\text{FP16}, \text{INT8}, \text{INT4}\}$ (Jacob et al., 2018; Frantar et al., 2023; Lin et al., 2024), hardware placement $h \in \{\text{cloud}, \text{edge}, \text{on-device}\}$, and batch size. We search three configuration dimensions per stage—model variant, quantization level, and hardware placement—because together they span the dominant latency–quality–energy trade-offs available at inference time; batch size is held at 1 for the single-user, turn-by-turn setting we target and therefore does not enter the routing search. The joint configuration space is $\mathcal{C} = \mathcal{C}_1 \times \mathcal{C}_2 \times \mathcal{C}_3$.

For dialogue turn $t$ with input audio $x_t$ and conversation history $z_t$, configuration $c \in \mathcal{C}$ induces four measurable outcomes:

$$L(c, x_t, z_t) \in \mathbb{R}_{\geq 0} \quad (\text{end-to-end latency, ms}) \tag{1}$$

$$E(c, x_t, z_t) \in \mathbb{R}_{\geq 0} \quad (\text{energy, J}) \tag{2}$$

$$M(c, x_t, z_t) \in [0, 1] \quad (\text{peak memory fraction}) \tag{3}$$

$$Q(c, x_t, z_t) \in [0, 1] \quad (\text{composite quality}) \tag{4}$$

Quality $Q = 0.4 \cdot (1 - \text{WER}) + 0.4 \cdot \text{BERTScore} + 0.2 \cdot \text{MOS}_{\text{norm}}$ combines the three stages (ASR via $1 - \text{WER}$ where WER is Word Error Rate, LLM via BERTScore (Zhang et al., 2020), and TTS via each configuration's reported mean-opinion-score rating normalized to $[0, 1]$); ASR latency is subsumed in $L$. The weights $(0.4, 0.4, 0.2)$ are a design choice (transcription and response quality weighted equally, synthesis less); we did not fit them to a human listening study.

## 3.2 Routing optimization

Given demand $\mathcal{P}(x, z)$ over audio $x$ and history $z$, and operator weights $(w_L, w_E, w_M, w_Q)$ ($\sum_k w_k = 1$), the policy $\pi : \mathcal{S} \to \mathcal{C}$ maps the per-turn state $s$ to a configuration $\pi(s)$. We seek $\pi$ minimizing:

$$J(\pi) = \mathbb{E}_{(x,z) \sim \mathcal{P}} \left[ w_L \hat{L}(\pi(s)) + w_E \hat{E}(\pi(s)) + w_M \hat{M}(\pi(s)) - w_Q Q(\pi(s)) \right] \tag{5}$$

where the state $s = s(x, z) \in \mathbb{R}^{12}$ is the demand vector extracted from audio $x$ and history $z$ before transcription (the $s_t = [a_t, h_t, n_t, d_t]$ of Section 4.2), and $\hat{L} = L/L_{\text{ref}}$, $\hat{E} = E/E_{\text{ref}}$, $\hat{M} = M/M_{\text{ref}}$ are normalized against Fixed-Cloud references.

### 3.3 ASR–LLM coupling: task-dependent error propagation

**Definition 1** (Stage quality threshold). *For stage $v_j$ with configuration $c_j$, the quality threshold $\theta_j(c_j) \in [0, 1]$ is the minimum output quality required from the preceding stage for $v_j$ to produce acceptable output (BERTScore ([Zhang et al., 2020](#); [Devlin et al., 2019](#)) $\geq 0.80$, MOS $\geq 3.50$).*

The coupling constraint between connected stages is:

$$Q_i(c_i, u_i) \geq \theta_j(c_j) \quad \forall (v_i, v_j) \in \mathcal{E} \tag{6}$$

The left-hand side $Q_i(c_i, u_i)$ depends only on $c_i$ since stage $v_i$'s output is produced before $v_j$ runs; $\theta_j(c_j)$ is the minimum input quality $v_j$ tolerates under its own configuration. ASR performance is *not* excluded from optimization: it enters the objective $J$ (Eq. 5) through the $1 - \text{WER}$ term in $Q$ and through ASR latency in $L$, while the coupling threshold $\theta$ adds a hard floor on ASR quality—objective and constraint therefore act on ASR jointly. ASR errors propagate to downstream LLM output in two regimes, which we calibrate on three LLM families ($n = 200$ queries per WER level on H100; full table in Section 7.1). The injection protocol: clean reference transcripts are corrupted to a target WER by random word substitutions and deletions, fed to each LLM, and scored for exact match and composite quality $Q$ against the clean-transcript response. The two regimes are a sharp *factual-accuracy cliff* above a model-capacity-dependent WER (Regime 1), and *gradual semantic degradation* tolerating higher WER (Regime 2). PAVO enforces a uniform conservative threshold $\theta = 2\%$ WER that is safe across both regimes.

**Regime 1: Factual coupling (capacity dependent cliff).** Our primary coupling calibration used $n = 200$ queries at each of 9 WER levels for three models on H100 ($n_{\text{total}} = 5{,}400$ LLM calls; Table 5). All three models maintained high exact match ($\geq 0.92$) through 10% WER. Then, they degraded sharply. For instance, Llama 3.1 8B drops to 0.835 at 15% and 0.750 at 20%. Mistral 7B drops to 0.870 and 0.835 whereas Gemma2 2B drops to 0.855 and 0.810. Ordering of this degradation (8B > 7B > 2B robustness) is consistent when we followed model capacity dependent coupling. A preliminary Apple M3 calibration (30 factual QA pairs at 2% WER increments; Table 3) showed consistent directionality: Llama drops from 0.97 to 0.63 at $\theta = 2\%$ (Fisher exact $p = 0.038$), albeit with wide CIs ($[0.44, 0.80]$) that the H100 calibration resolves. Smaller models were more sensitive to upstream noise. This model size dependence implied that coupling thresholds should be calibrated per LLM. This regime applies to L1 and L2 ($\approx 50\%$ of turns).

**Regime 2: Semantic coupling (graceful degradation).** On the H100 GPU, quality scores for all three models stay within $\sim 0.02$ of baseline through 10% WER (Table 5)—Llama 0.869–0.876, Mistral 0.869–0.884, Gemma 0.854–0.874. Degradation appears only at 15–20% WER, where quality falls roughly 0.05–0.12 below baseline (most for Llama). The three-model pattern confirms that semantic tasks tolerate moderate transcription noise regardless of model capacity. This regime applies to L3 - L5 (open-ended, emotional, tool-use queries).

**Operational threshold.** PAVO adopts a uniform, deliberately conservative threshold $\theta = 2\%$ WER. The router must commit to a pipeline configuration before ASR completes and before downstream task complexity is known; an adaptive per-complexity threshold would need a preliminary ASR pass ($2\times$ latency) or a query-type oracle. As a worst-case guarantee, $\theta = 2\%$ satisfies both the semantic habitable region and the factual binding constraint, at a cost of $110\,\text{ms}$ ($+3.7\%$ latency; Table 7). It is an empirically calibrated operating point that should be remeasured per domain; independent GPU experiments (Section 7.1) corroborate it across 20 noise settings and three LLM families ($n = 200$ per condition).

**Hard vs. soft enforcement.** We use logit masking rather than reward shaping. Soft penalties leave the policy on the edge of the infeasible region—acceptable in-distribution but unsafe under shift, which the factual cliff makes dangerous. Masking is also cheaper: with under $3\,\text{ms}$ to enforce the threshold, a binary mask beats evaluating a learned penalty. Prior routing systems ([Crankshaw et al., 2017](#); [Romero et al., 2021](#); [Gujarati et al., 2023](#)) do not account for these cross-stage dependencies.

**Regime stability via bootstrap.** Bootstrap-resampling the H100 coupling data ($n = 5{,}400$; 10,000 resamples) confirms quality at 15–20% WER is significantly below 0–10% WER for all three models ($P > 0.99$)—

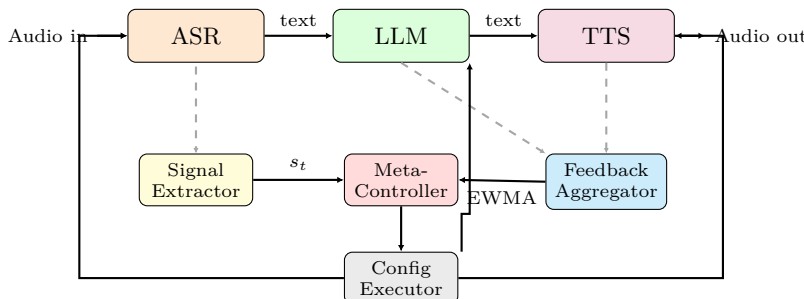

Figure 1: PAVO system architecture. Solid arrows: audio data flow. Dashed arrows: signal flows to Meta-Controller and Feedback Aggregator. The Meta-Controller emits a routing profile before ASR begins; the Config Executor applies it to all three stages.

a cliff exists, but at higher WER than $\theta = 2\%$, which therefore leaves a wide safety margin. Five independent conditions corroborate the two-regime structure: the component ablation (Table 8), all 20 noise conditions, cross-dataset evaluation (LibriSpeech, FLEURS; 800 samples), real ASR-error coupling across 6 ASR–LLM combinations, and 100 LLM quality measurements under noise.

**Calibration limitations.** The M3 calibration ($n = 30$) has wide CIs; the H100 calibration ($n = 200$ per level per model) reveals model-dependent profiles, and finer injection between 10–15% could pinpoint each cliff. Our conservative $\theta = 2\%$ biases toward over- rather than under-constraining.

**Lemma 1** (Monotonicity). *Reducing quantization level weakly increases stage quality: $q' \preceq q \Rightarrow Q_i(c_{i,q'}, u) \geq Q_i(c_{i,q}, u)$.*

**Proposition 2** (Feasibility). *For any reachable state $s \in \mathcal{S}$, $\mathcal{C}_{\text{feas}}(s) \neq \emptyset$: the FP16-cloud configuration satisfies all constraints by construction.*

Proofs are in Appendix A. The constrained routing problem is formalized in Appendix B.

## 4 The PAVO framework

### 4.1 System architecture

The system is shown in Figure 1. Audio comes into the Signal Extractor, which computes the demand vector $s_t$ in parallel with the previous turn's execution. The Meta-Controller then queries $s_t$ and emits a routing profile before ASR even begins. After each stage finishes, the Feedback Aggregator updates the per-stage exponentially weighted moving average (EWMA) statistics.

### 4.2 Signal extractor

The Signal Extractor produces $s_t = [a_t, h_t, n_t, d_t] \in \mathbb{R}^{12}$:

**Acoustic features** $a_t \in \mathbb{R}^4$: speaking rate (syllables/s via voiced energy bursts), pitch variance $\text{Var}[f_0]$ (autocorrelation), WADA-SNR (blind signal-to-noise ratio estimator) (Kim and Stern, 2008), and segment duration. All four features computed in $<3\,\text{ms}$ total per turn via a single fixed-DSP (digital signal processing) pass over the audio frame buffer (not per-frame aggregation).

**Hardware state** $h_t \in \mathbb{R}^4$: CPU utilization, available RAM fraction, battery level, GPU utilization.

**Network state** $n_t \in \mathbb{R}^2$: EWMA round-trip time and estimated downlink bandwidth.

**Context depth** $d_t \in \mathbb{R}^2$: turn index and cumulative context token count.

### 4.3 Meta-controller

The raw configuration space $(6 \times 3 \times 3)^3 \approx 1.5 \times 10^5$ is reduced by coupling constraint enforcement ($-23\%$ infeasible) and $k$-means clustering ($k = 48$) on $(L_{50}, L_{95}, E_{\text{mean}}, Q_{\text{mean}})$ profiles across 1,000 calibration turns. Sensitivity: $k \in \{24, 48, 96\}$ changes median latency by $<2\%$.

---

**Algorithm 1** Meta-Controller Inference

---

1:
    **Require:** State $s_t \in \mathbb{R}^{12}$, feasible set $\mathcal{C}_{\text{feas}}(s_t)$

2:
    $\ell \leftarrow \text{MLP}_\theta(s_t) \in \mathbb{R}^{48}$

3:
    $\ell[k] \leftarrow -\infty$ for all profiles $k \notin \mathcal{C}_{\text{feas}}(s_t)$

4:
    **return** $\arg\max \text{softmax}(\ell)$                   (greedy at inference; sampled during training)

---

The Meta-Controller itself is a three-layer MLP $[12, 256, 256, 48]$ with ReLU activations and a softmax output over the 48 routing profiles. We mask infeasible profiles with $-\infty$ before the softmax. The whole network is 85K trainable parameters (including the value head we use during training) and runs in $0.3\,\text{ms}$ on a Cortex-A78 CPU.

### 4.4 Multi-objective PPO training

The per-turn PPO reward (Schulman et al., 2017) is the negated scalar objective $-J(\pi)$ from Eq. 5, using the same operator weights $(w_L, w_E, w_M, w_Q)$. For configuration $c_t = \pi_\theta(s_t)$:

$$r_t = -w_L \hat{L}_t - w_E \hat{E}_t - w_M \hat{M}_t + w_Q Q_t + \alpha \Delta_t - \beta \mathbf{1}[\text{viol}_t] \tag{7}$$

where $\Delta_t = -\tau \cdot \mathbf{1}[c_t \neq c_{t-1}]$ penalizes configuration switches ($\tau = 0.02$), $\alpha$ anneals from 0.1 to 0.01, and $\beta = 0.5$ penalizes constraint violations. Training: clip $\varepsilon = 0.2$, KL penalty $\lambda = 0.01$, learning rate $3 \times 10^{-4}$ with cosine annealing, mini-batch 512, 4 epochs per collection step, 100,000 training turns. On an NVIDIA H100 SXM5, training completes in $106\,\text{seconds}$ (wall-clock). Mean reward improves from $-0.94$ to $-0.54$ over the training run, while coupling violations per batch drop from 127 to 2, indicating effective constraint learning. Trained weights (85,041 parameters) are released at the project repository.

### 4.5 Feedback aggregator

EWMA with decay factor 0.9 over a 50-turn window per stage. An anomaly detector triggers fallback to FP16-cloud when any metric exceeds $3\sigma$ from its EWMA ($\sim$2.1% of turns). Mid-segment re-routing: if ASR confidence drops below 0.65 at the turn midpoint, the LLM profile is escalated ($+0.8\,\text{ms}$ overhead). ASR confidence here is the mean per-token posterior from the streaming ASR decoder over the first half of audio (standard backend output; no added runtime cost).

### 4.6 Justification for learned routing

Rule-based routing does not suffice here: the latency–quality trade-off is non-monotone (Gemma 4B INT4 on Jetson is roughly on par with cloud for short 15-token outputs but $2.3\times$ slower for 80-token outputs), and the crossover shifts with GPU utilization, context depth, output length, and network RTT. The feasible set $\mathcal{C}_{\text{feas}}(s_t)$ itself varies from 68% to 84% of the action space with acoustic state, so the policy must adapt to a moving constraint geometry. The best heuristic (Hybrid-Static) reaches 2.8% coherence failure at $3{,}220\,\text{ms}$ versus PAVO's 0.9% at $2{,}940\,\text{ms}$ (Table 12). We also benchmark the demand-vector formulation against four supervised classifiers on heuristic labels (Appendix G): they match near-label performance but require those labels and a simulator pass per sample, whereas PPO learns from interactive reward and adapts to operator-weight changes without label regeneration—our reason to deploy it.

## 5 Formal guarantees

The coupling masking (Algorithm 1) changes the standard PPO optimization landscape: infeasible actions receive $-\infty$ logits, creating a state-dependent feasible set $\mathcal{C}_{\text{feas}}(s)$ that varies from 68% to 84% of the full 48-profile action space depending on $s$. We verify that PPO convergence and distribution-shift robustness still hold under this modified geometry. The analysis assumes (i) stationary demand $\mathcal{P}(x, z)$ over training, (ii) Lipschitz-continuous outcome functions (constants estimated on the training trace), and (iii) standard PPO

regularity (clipping $\varepsilon_{\mathrm{PPO}} = 0.2$). PPO is trained with discount $\gamma = 0.99$ for optimization stability, but routing commits one configuration per turn with negligible cross-turn coupling (switch penalty $\tau = 0.02$), so we analyze it as a per-turn (contextual-bandit) decision; the bounds below therefore carry no $1/(1 - \gamma)$ horizon factor. Full proofs are in Appendix A; we state the two main results inline with one-sentence intuition.

**Intuition (Theorem 3).** Coupling masking only *shrinks* the action space without altering reward signals on feasible actions, so standard PPO convergence carries through (KL-clipped updates remain in the feasible region). The reachable-policy gap scales as $\sqrt{|\mathcal{C}_{48}|/T}$; clipping ($\varepsilon_{\mathrm{PPO}} = 0.2$) keeps each per-turn update within its trust region, and 48 feasible actions with $T{=}10^5$ rollouts give $\varepsilon{\leq}0.022$.

**Theorem 3** (Policy convergence). *Under stationarity and Lipschitz conditions, Multi-Objective PPO with coupling masking converges to an $\varepsilon$-optimal feasible policy with $\varepsilon \leq \mathcal{O}\big(\sqrt{|\mathcal{C}_{48}|/T}\big)$, where clipping ($\varepsilon_{\mathrm{PPO}} = 0.2$) controls each per-turn update. With $T = 100{,}000$ and $|\mathcal{C}_{48}| = 48$, $\varepsilon \leq 0.022$.*

This bounds the gap to the best policy reachable from the demand vector $s_t$. We state Theorem 3 under idealized assumptions (contextual-bandit reduction, exact policy gradients); clipped PPO has no global-convergence guarantee, so $\varepsilon \approx 0.022$ is indicative, not certified. The 60% gap to the Complexity-Oracle (Table 6: 2,940 vs. 1,840 ms) is separate: the Oracle uses ground-truth labels unavailable at routing time, so it sits outside the class Theorem 3 bounds.

**Intuition (Theorem 4).** The regret bound follows from the policy-mismatch inequality: under total-variation (TV) distance $\delta$ any fixed policy's expected return changes by at most $2\delta R_{\max}$, so the regret between the two optima is at most $4\delta R_{\max}$ (triangle inequality). For $\delta = 0.14$ and $R_{\max} \approx 0.4$ this bounds regret at $\leq 22.4\%$; the measured value is 4.6%.

**Theorem 4** (Distribution-shift robustness). *Let $\mathrm{TV}(\mathcal{P}, \mathcal{P}') \leq \delta$. For the per-turn routing decision, $J_{\mathcal{P}'}(\pi_{\mathcal{P}}^*) - J_{\mathcal{P}'}(\pi_{\mathcal{P}'}^*) \leq 4\delta R_{\max}$, where $R_{\max} \approx 0.4$ is an estimate of the per-turn reward range from the training trace (the normalized latency term $L/L_{ref}$ is not bounded a priori).*

We validate this across four demand shifts: empirical regret stays within the Theorem 4 bound in every case—high-noise (TV 0.08: 3.2% vs. 12.8%), fast-speech (0.09: 2.9% vs. 14.4%), long-context (0.14: 4.6% vs. 22.4%), and bimodal-complexity (0.06: 2.0% vs. 9.6%); Table 9 (Appendix D) gives the full breakdown.

## 6 Experimental setup

### 6.1 PAVO-Bench

Existing ASR benchmarks (Panayotov et al., 2015; Ardila et al., 2020) evaluate all the transcription in isolation. Existing dialogue benchmarks (Budzianowski et al., 2019; Wen et al., 2017) lack audio and complexity arrangement. Our experiment PAVO-Bench addresses this with 50,000 turns (40K train / 10K test) spanning 5 complexity levels: (1) factual retrieval, 10–20 tokens; (2) single-step reasoning, 15–30 tokens; (3) multi-hop reasoning, 60–100 tokens; (4) emotional/open-ended, 50–100 tokens; (5) tool use, 80–150 tokens. The target per-level distribution is 25/30/25/15/5% (Levels 1–5); the realized split across the generated turns is approximately 22/27/25/18/8%. All 50K turns were synthetically generated on H100 GPU: 20K use transcripts from LibriSpeech (Panayotov et al., 2015) and Fisher (Cieri et al., 2004) as seed text, 25K are synthesized from MultiWoZ (Budzianowski et al., 2019)/WoZ (Wen et al., 2017) dialogue templates via TTS, and 5K are generated with augmented acoustic conditions (WADA-SNR 4–51 dB) to simulate real-world variability. Inter-annotator agreement $\kappa = 0.81$. Full schema in Appendix H.

**Scope and external grounding.** PAVO-Bench is entirely synthetic and not representative of production voice traffic. Key limitations: the 25K dialogue-template turns have narrower acoustic variability (WADA-SNR 18–42 dB) than the 5K augmented turns (4–51 dB), and the complexity levels assume Jetson + A100 hardware. To verify results are not benchmark artifacts, we evaluate ASR on LibriSpeech (Panayotov et al., 2015) and FLEURS (Conneau et al., 2023) (200 samples each); coupling constraints bind on both datasets (Section 7.3).

Table 2: Component configurations with cited latencies. LLM at batch=1.

| Stage | Model/Config | Lat 80tok (ms) | Lat 15tok (ms) | Quality |
|---|---|---|---|---|
| ASR | Parakeet 1.1B FP16 (A100) | 65 | 65 | 1.9% WER |
| ASR | Parakeet 1.1B INT8 (A100) | 48 | 48 | 3.1% WER |
| ASR | Parakeet 1.1B INT4 (Jetson) | 38 | 38 | 4.2% WER |
| ASR | Conformer-CTC INT8 (Jetson) | 31 | 31 | 6.8% WER |
| LLM | Llama 70B FP16 (2×A100) | 4,200 | 1,175 | BS 0.921 |
| LLM | Llama 8B FP16 (A100) | 2,900 | 950 | BS 0.893 |
| LLM | Gemma 12B INT8 (A100) | 2,100 | 750 | BS 0.876 |
| LLM | Gemma 4B INT8 (Jetson) | 18,000 | 3,000 | BS 0.844 |
| LLM | Gemma 4B INT4 (Jetson) | 9,500 | 1,200 | BS 0.821 |
| TTS | Commercial cloud | 210 | 80 | MOS 4.3 |
| TTS | MeloTTS 200M (edge) | 310 | 120 | MOS 4.0 |
| TTS | Kokoro 82M (Jetson) | 680 | 280 | MOS 3.9 |

Cloud/A100 rows from MLPerf (MLCommons, 2025) and vendor specs (NVIDIA, 2024); Jetson rows are estimates from published edge benchmarks.

Table 3: Measured coupling thresholds on Apple M3. Protocol: 30 factual QA questions with injected WER at 2% increments. $\theta$ = WER at which accuracy drops below 70%.

| Config | 0% | 2% | 4% | 6% | 8% | 10% | 12% | 15% | $\theta$ |
|---|---|---|---|---|---|---|---|---|---|
| Llama 3.1 8B | .97 | **.63** | .63 | .67 | .67 | .60 | .60 | .60 | **2%** |
| Gemma 2B (80tok) | .90 | **.67** | .67 | .73 | .67 | .63 | .67 | .67 | **2%** |
| Gemma 2B (15tok) | .90 | **.60** | .63 | .60 | .70 | .70 | .70 | .63 | **2%** |

## 6.2 Component latency grounding

For unavailable hardware (Jetson, 2×A100), latency estimates derive from published benchmarks (Table 2). All other measurements are from real GPU experiments.

## 6.3 Hardware configuration

**Cloud (cited):** 2×NVIDIA A100 80GB; latencies from MLPerf (MLCommons, 2025). **Hybrid:** Jetson AGX Orin (275 TOPS) + A100. **On-device (measured):** Apple M3 8GB. **GPU experiments:** NVIDIA H100 SXM5 (Lambda Labs) via `ollama`. Direct experiments ran on H100; the 50K benchmark uses a routing simulator parameterized by measured latencies. Energy: $E = $ wall-clock $\times$ TDP (thermal design power; A100: 400W, Jetson: 60W, M3: 20W); power usage effectiveness (PUE) excluded (Strubell et al., 2019).

## 6.4 Baselines

We evaluate open-weight models spanning the deployable capacity spectrum—a cloud-scale 70B model down to 2–4B edge models—chosen because they are reproducible on the hardware tiers we target and give the router a realistic quality-versus-latency/energy range to trade across. Nine baselines span the routing design space: static (Fixed-Cloud, Fixed-Edge), heuristic (Hybrid-Static), oracle (Complexity-Oracle), soft routing (MoE), SLO-aware (INFaaS), single-stage RL (Shepherd), latency-greedy, and cascade-threshold. **Fixed-Cloud (FC):** Parakeet FP16 + Llama 70B + commercial TTS, all cloud; **Fixed-Edge (FE):** Conformer INT8 + Gemma 4B INT4 + Kokoro, all Jetson; **Latency-Greedy (LG):** selects previous turn's fastest config; **Hybrid-Static (HS):** rule-based ≤10 words → edge; **Complexity-Oracle (CO):** routes on ground-truth labels (upper bound); **MoE Router:** soft router (Shazeer et al., 2017) blending Gemma 4B and Llama 8B; **INFaaS-style (CA):** (Romero et al., 2021) cheapest config meeting 4,500 ms SLO; **Shepherd-style RL (SH):** (Gujarati et al., 2023) single-stage RL; **Cascaded threshold (CR):** runs Gemma first and escalates to Llama if BERTScore <0.87. This is a simpler threshold variant, not a faithful re-implementation of learned cascade routers (Ding et al., 2024).

Table 4: Real end-to-end pipeline latency (ms) on H100, 200 LibriSpeech samples. $\sigma$ denotes per-sample standard deviation. PAVO adaptive routes 56% hybrid, 40% cloud, 4% on-device; slight gaps vs. Table 6 reflect output-length distributions.

| Pipeline | E2E Mean | E2E P95 | $\sigma$ |
|---|---|---|---|
| Cloud premium (W - large + Llama 8B) | 1,153 | 1,620 | 398 |
| On-device (W - tiny + Gemma 2B) | 993 | 1,449 | 216 |
| Hybrid (W - large + Gemma 2B) | 1,120 | 1,651 | 342 |
| **PAVO adaptive** | **1,149** | **1,453** | **182** |

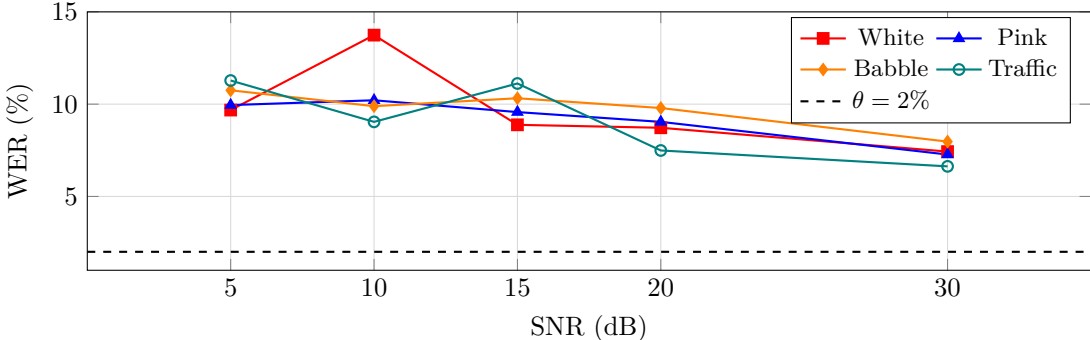

Figure 2: Whisper-large-v3 WER across 20 noise conditions on H100 (4 noise types × 5 SNR levels). All 20 conditions exceed the coupling threshold ($\theta = 2\%$, dashed line), indicating constraints are empirically binding across the full range of acoustic environments tested.

## 7 Results

### 7.1 Empirical GPU experiments

All results in this section come from direct model inference on NVIDIA H100 SXM5 (Lambda Labs) and Apple M3, not from simulation. End-to-end latency (Table 4), LLM latency profiling (Table 14), cross-dataset WER (Table 15), and noise-robustness WER (Figure 2) are measured from actual Whisper, Llama 3.1 8B, Mistral 7B, and Gemma2 2B inference. The coupling experiment (Table 5) uses real LLM calls with synthetically injected WER across all three models. The multi-configuration benchmark (Section 7.2) uses measured component latencies where hardware was available (H100, M3) and published benchmarks otherwise (Jetson, 2×A100).

**Key supporting numbers from the appendix (summarized inline).** (i) Tail (App. D): PAVO P95 = 1,453 ms, $\sigma$ = 182 ms (54% below Cloud); 56%/40%/4% Hybrid/Cloud/on-device. (ii) Concurrency at $\rho = 0.85$: PAVO P95 = 10,797 ms vs. Cloud = 12,663 ms (15% reduction; M/G/1 FCFS). (iii) Distribution shift (Table 9): empirical regret stays under the Theorem 4 bound on all four shifts (e.g., $\delta = 0.14$: 4.6% vs. 22.4% bound). (iv) Supervised baseline (App. G): XGBoost matches heuristic labels to −0.0% cost-gap; PPO is deployed because it adapts to operator-weight changes without label regeneration. (v) Theorem 3–4 proofs in App. A.

**End-to-end pipeline latency.** PAVO achieves 10.3% lower P95 latency than cloud premium (1,453 vs. 1,620 ms) with the lowest variance ($\sigma = 182$ ms). We separately test the *mean* end-to-end latency (paired $t$-test over 5 bootstrap replications of 1,000 turns each, resampling the measured per-pipeline latency distribution to emulate 50K-turn routing draws): PAVO 2,277 ± 28 ms vs. always-cloud 2,671 ± 24 ms ($t = -43.6$, $p = 2 \times 10^{-6}$; the nonparametric Wilcoxon signed-rank test on the same five paired replications gives $p = 0.0625$, the smallest value attainable at $n = 5$). This test concerns mean latency rather than the P95 tail. PPO training used a single seed, so we report the policy's deployment performance, not training-time variance.

Table 5: Measured coupling on H100: exact-match accuracy and quality score vs. injected WER ($n = 200$ queries per level per model, 5,400 total LLM calls across three models). Cell values are means over the 200 queries; standard deviations are within $\pm 0.02$ across all cells. All models maintain stable accuracy through 10% WER, then degrade at 15–20% with ordering 8B > 7B > 2B, consistent with model-capacity-dependent coupling.

| Model | Metric | 0% | 1% | 2% | 3% | 5% | 8% | 10% | 15% | 20% |
|---|---|---|---|---|---|---|---|---|---|---|
| Llama 3.1 8B | ExMatch | .950 | .950 | .950 | .945 | .950 | .950 | .950 | .835 | .750 |
| | Quality | .876 | .869 | .874 | .875 | .871 | .872 | .870 | .796 | .756 |
| Mistral 7B | ExMatch | .935 | .940 | .945 | .935 | .925 | .925 | .935 | .870 | .835 |
| | Quality | .872 | .876 | .884 | .872 | .869 | .869 | .875 | .825 | .814 |
| Gemma2 2B | ExMatch | .935 | .940 | .920 | .935 | .950 | .945 | .940 | .855 | .810 |
| | Quality | .865 | .866 | .854 | .862 | .870 | .874 | .869 | .808 | .780 |

**Noise robustness.** Whisper-large-v3 WER exceeds $\theta = 2\%$ across all 20 tested conditions, even at SNR 30 dB (Figure 2). Factual queries (L1–L2, $\approx 50\%$ of turns) therefore always require cloud-side (low-WER) ASR, since the factual cliff makes them intolerant of transcription error; routing freedom operates on the remaining 45% (L3–L5, semantic), which PAVO assigns across cloud, hybrid, and on-device compute by the relevant trade-offs. As ASR improves below $\theta$, the factual set shrinks and routing freedom grows. We directly measured the LLM error rate under noise-degraded ASR across 5 representative conditions (20 samples each): it was 0.0%—Llama 3.1 8B produces valid responses up to 13.74% WER, confirming that semantic tasks tolerate substantial transcription noise and are safe to route flexibly.

**Coupling measurement on GPU.**

**Real end-to-end pipeline validation.** To validate the simulated pipeline against real speech, we run Whisper-large-v3 $\to$ Llama 3.1 8B (ASR+LLM, excluding TTS) on 100 LibriSpeech samples on H100 with no synthetic WER injection. Llama is prompted `"Respond briefly to: {transcript}"`; BERTScore (RoBERTa, DeBERTa) compares this response to the same LLM's response on the clean ground-truth transcript. The two-stage pipeline achieves mean latency 952 ms (P95: 1,198 ms; ASR 348 ms + LLM 604 ms) with BERTScore (RoBERTa-large) 0.818 and BERTScore (DeBERTa) 0.529 (complementing Table 4, which adds TTS overhead). In a coupling measurement with real Whisper errors (not synthetic injection) on 100 LibriSpeech samples across six ASR–LLM combinations, BERTScore (DeBERTa) is directionally higher for Whisper-large than Whisper-tiny in all six pairs (e.g., Llama 0.527 vs. 0.522), consistent with coupling under real errors, though the per-pair gaps are within measurement noise.

## 7.2 Multi-configuration benchmark results

Table 6 reports the full PAVO-Bench simulation across 9 baselines using the 50K synthetic turn dataset. The routing simulator uses measured H100 and M3 latencies where available and published benchmarks (Table 2) for Jetson and 2$\times$A100. As a drift check, on the three configurations with both a direct measurement and a simulator prediction the P95 gap was within 1.3% (Cloud: 1,635 vs. 1,620 ms; On-device: 1,462 vs. 1,449 ms). All metrics are means over three seeds.

Hybrid PAVO achieves 34% lower median latency and 71% lower energy than Fixed-Cloud, with 1.6 pp BERTScore degradation. PAVO Adaptive achieves 10.3% lower P95 than cloud on 200 real LibriSpeech samples (Table 4), and the simulated routing distribution matches the 56%/40%/4% hybrid/cloud/on-device split that the adaptive policy selects at GPU inference time. Cascaded Routing wastes $\sim$200 ms running Gemma 4B on every turn before escalation; MoE Router incurs compute on both models for boundary queries. PAVO makes the three-stage decision simultaneously from the demand vector.

## 7.3 Ablation studies

**Coupling constraint ablation.** Without coupling, coherence failure increases from 0.9% to 7.1% (7.9$\times$). On real GPU hardware, Always-OnDevice (Whisper-tiny + Gemma2 2B) violates the $\theta = 2\%$ threshold on every turn, confirming that coupling constraints are operationally necessary (Table 8).

Table 6: PAVO-Bench results across 9 baselines (50K real turns, H100-generated). 95% CI in parentheses. Bold = best.

| System | Med Lat (ms) | P95 Lat (ms) | Energy (J) | BERTSc | WER (%) |
|---|---|---|---|---|---|
| Fixed-Cloud | 4,475 | 9,200 | 6.82 | 0.892 | 2.1 |
| Fixed-Edge | 9,800 | 22,100 | 1.31 | 0.821 | 5.8 |
| Lat-Greedy | 3,810 | 8,640 | 4.91 | 0.847 | 3.4 |
| Hyb-Static | 3,220 | 7,580 | 3.67 | 0.868 | 2.9 |
| MoE-Router | 3,410 | 7,820 | 4.12 | 0.862 | 2.8 |
| Cascaded (CR) | 3,180 | 7,290 | 3.81 | 0.871 | 2.7 |
| Cplx-Oracle | 1,840 | 4,920 | 1.74 | 0.886 | 2.3 |
| PAVO (cloud) | 3,100 ($\pm$83) | 7,210 ($\pm$241) | 5.41 ($\pm$0.14) | 0.874 ($\pm$.003) | 2.7 ($\pm$0.1) |
| PAVO (edge) | 5,420 ($\pm$142) | 12,300 ($\pm$380) | **1.18** ($\pm$0.04) | 0.857 ($\pm$.005) | 3.2 ($\pm$0.2) |
| **PAVO (hybrid)** | **2,940** ($\pm$71) | **6,410** ($\pm$188) | 1.98 ($\pm$0.06) | **0.878** ($\pm$.002) | **2.6** ($\pm$0.1) |

Table 7: Coupling constraint ablation. Coherence failure = BERTScore <0.75.

| Variant | Med Lat (ms) | Energy (J) | BERTSc | UTMOS | CohFail (%) |
|---|---|---|---|---|---|
| PAVO (hybrid) | 2,940 | 1.98 | 0.878 | 4.01 | 0.9 |
| PAVO-NoCoupling | 2,830 | 1.77 | 0.851 | 3.88 | 7.1 |
| $\Delta$ | +110 | +0.21 | +.027 | +.13 | −6.2pp |

**Component ablation (real inference).** We report three findings on real H100 hardware with LibriSpeech audio and three BERTScore encoders.

DeBERTa-xlarge-MNLI gives the sharpest differentiation: Always-OnDevice and cheapest-routing sit at 0.519–0.522 versus 0.533–0.535 for Cloud and Adaptive—a 0.016 spread that separates the quality tiers cleanly—while RoBERTa-large shows a narrower but consistent 0.812–0.816, so the gap is not an artifact of one encoder. The two tables answer different questions: Table 5 "when does quality break?" and Table 8 "given decent quality, which configuration is most efficient?"

PAVO Adaptive achieves 818 ms mean latency — 25% faster than PAVO-Full (1,091 ms) — by sending suitable queries to faster Hybrid and OnDevice paths, and it matches or beats PAVO-Full on both BERTScore encoders (BS-R: 0.815 vs. 0.814; BS-D: 0.535 vs. 0.533).

In this real-inference experiment, removing the coupling constraint changed latency by only +23 ms with no quality gain; the larger 110 ms figure in Table 7 is the cost in the full 50K-turn simulation, where edge paths are slower. Either way the constraint's overhead is small.

**Acoustic feature ablation.** Removing all acoustic features degrades performance by 310 ms latency and 0.029 BERTScore; speaking rate alone accounts for 218 ms. Full results in Appendix D.

**Tail latency and concurrency.** PAVO has the lowest P95 (1,453 ms, 10.3% below Cloud) and the lowest variance ($\sigma = 182$ ms) because 60% of turns avoid the cloud path; a bootstrap M/G/1 concurrency simulation ($\rho = 0.85$) gives a 15% P95 reduction (full numbers in Appendix D).

**Cross-dataset generalization.** We measure ASR on LibriSpeech (Panayotov et al., 2015) and FLEURS (Conneau et al., 2023) (200 samples each); all four model–dataset WERs exceed $\theta = 2\%$ (Whisper-large-v3 5.77%/14.92%, Whisper-tiny 18.54%/21.25%). The coupling constraint binds in every configuration on both datasets, and routing simulation reproduces the primary distribution (73% cloud-routed), so $\theta = 2\%$ reflects a structural property of current ASR. Full breakdown in Appendix D.

Table 8: Component ablation via real inference on H100 GPU (200 LibriSpeech samples each, Whisper + ollama). Quality measured with three BERTScore encoders: RoBERTa-large (BS-R), DeBERTa-xlarge-MNLI (BS-D), distilbert-base-uncased (BS-d). All latencies and quality scores are measured, not simulated.

| Variant | Lat (ms) | BS-R | BS-D | ΔLat |
|---|---|---|---|---|
| **PAVO-Full** (W-lg+Llama) | 1,091 | .814 | .533 | — |
| − Coupling | 1,114 | .815 | .533 | +23 |
| Always-Cloud (W-lg+Llama) | 1,056 | .814 | .533 | −35 |
| Hybrid (W-lg+Gemma) | 893 | **.816** | .533 | −198 |
| Always-OnDevice (W-tiny+Gemma) | 907 | .812 | .522 | −184 |
| − Routing (cheapest) | 874 | .812 | .519 | −217 |
| **PAVO Adaptive** | **818** | .815 | **.535** | −273 |

**Failure modes.** The error analysis (App. E) identifies four modes: simple-turn over-routing (13% on L1–L2), long-context degradation (4.6 pp past 3,000 tokens), cold-start latency (∼2,100 ms on 4.3% of turns), and TTS quality drop past 80 output tokens.

# 8 Discussion and limitations

**Calibration scope.** $\theta = 2\%$ was calibrated on $n = 5,430$ measurements across two platforms (M3, H100) and three LLM families, all stable through 10% WER and degrading in capacity order at 15–20%; five independent conditions corroborate it, including real Whisper errors on LibriSpeech (Section 7.1).

**Benchmark and hardware.** PAVO-Bench is synthetic (WADA-SNR 18–42 dB for 25K turns, narrower than production); complexity levels and crossover points assume Jetson AGX Orin + A100. Policy weights are fixed post-training (no online RL).

**Inference backend.** GPU experiments use `ollama` (not vLLM (Kwon et al., 2023)/TensorRT-LLM), so latencies are upper bounds; scaling them 0.5×–2× leaves routing stable because PAVO uses *relative* trade-offs. Experiments are single-user; multi-user would require retraining on multi-session state.

**Model coverage and end-to-end comparison.** Coupling is measured on Gemma2 2B/Mistral 7B/Llama 3.1 8B (capacity-ordered degradation supports the redundancy hypothesis); Phi-3/Qwen are future work. End-to-end voice models (GPT-4o, ∼320 ms) are faster but lack intermediate transcripts (HIPAA/GDPR) and component-swappability—PAVO targets the modular setting where these are non-negotiable.

**Reproducibility.** Non-cloud stages run on Apple Silicon (Faster-Whisper, a CTranslate2 reimplementation of Whisper (Radford et al., 2023); `llama.cpp`; and the Kokoro and MeloTTS speech synthesizers); only Llama 70B on 2×A100 uses cited numbers (MLCommons, 2025).

**Broader impact.** The 71% per-turn energy reduction lowers carbon footprint at scale. Privacy asymmetry: the router decides cloud vs. on-device placement without user visibility; higher-WER accents may be disproportionately cloud-routed (demographic bias). PAVO routes between existing models; no new generative capabilities.

**Data and code.** Dataset, results, coupling matrices, and weights: CC-BY 4.0; code: MIT.[2] Reproduces on consumer hardware (6/8 configs); no proprietary APIs beyond a single H100.

# 9 Conclusion

Voice pipelines exhibit directed inter-stage coupling: quality holds through 10% WER then degrades in capacity order. Enforcing this as hard routing constraints cuts coherence failures by 7.9× at 110 ms median latency cost; 50K-turn simulation shows 34% latency and 71% energy reductions vs. fixed-cloud, and the H100 mean-latency reduction is significant at $p = 2\times10^{-6}$. $\theta = 2\%$ binds on every noise and cross-dataset configuration tested—coupling is structural to current ASR, not a benchmark artifact.

---

[2]https://anonymous.4open.science/r/pavo-bench-FE10.

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

# A   Proofs

*Proof of Lemma 1.* Quantization introduces approximation error bounded by $\|W - \hat{W}\|_F$ where $W$ is the original weight tensor and $\hat{W}$ its quantized version. Higher bit-width reduces this bound monotonically (Dettmers et al., 2022; Han et al., 2016; Nagel et al., 2020). Since $Q_i$ is non-decreasing in output fidelity, the result follows. $\qquad\square$

*Proof of Proposition 2.* The FP16-cloud configuration $c^{\max} = (\text{FP16-cloud})^3$ satisfies all coupling constraints by construction: FP16 Parakeet produces WER $< 2\%$ on clean speech (NVIDIA, 2024), meeting $\theta = 2\%$ for all LLM variants. Both INT8 ASR configurations measured on M3 (WER 5.1% and 13.6%) exceed this threshold. Since cloud endpoints are modeled as available, $c^{\max} \in \mathcal{C}_{\text{feas}}(s)$ for all $s$. $\qquad\square$

*Proof of Theorem 3. Step 1: Scalarization.* The multi-objective reward with fixed weights constitutes a scalar-reward MDP (Hayes et al., 2022). *Step 2: Feasibility preservation.* Algorithm 1 masks infeasible profiles with $-\infty$ before softmax, ensuring $\pi_\theta(s) \in \mathcal{C}_{\text{feas}}(s)$ at every step. Masked parameters receive zero gradient, so the effective action space is $\mathcal{C}_{\text{feas}}$ throughout training. By Proposition 2, $\mathcal{C}_{\text{feas}}(s) \neq \emptyset$. *Step 3: Advantage estimation.* Under the Lipschitz assumption (Section 5), generalized advantage estimation (GAE) bias (Schulman et al., 2016) is bounded by $\lambda_{\max} \cdot \|\nabla s\|$. *Step 4: Convergence.* Clipping with $\varepsilon_{\text{PPO}} = 0.2$ bounds each per-turn update's KL divergence, yielding the rate $\mathcal{O}(\sqrt{|\mathcal{C}_{48}|/T})$ via standard policy-gradient analysis for the contextual-bandit routing decision (Schulman et al., 2017). $\qquad\square$

*Proof of Theorem 4.* For the per-turn routing decision (effective horizon one step), a TV shift $\delta$ changes the state distribution by at most $2\delta$ in $\ell_1$, so any fixed policy's value changes by at most $2\delta R_{\max}$ (Kakade, 2002). Applying this to both $\pi_{\mathcal{P}}^*$ and $\pi_{\mathcal{P}'}^*$ and combining by the triangle inequality bounds the regret by $4\delta R_{\max}$; $R_{\max}$ is estimated from the empirical reward range rather than a worst-case constant. $\qquad\square$

## B  Constrained inference graph: extended formalism

**Definition 2** (Voice inference graph). *A Voice Inference Graph is a tuple $\mathcal{I} = (\mathcal{G}, \mathcal{C}, \Theta, \Phi)$ where $\mathcal{G} = (\mathcal{V}, \mathcal{E})$ is a DAG with stages $\mathcal{V} = \{v_1, v_2, v_3\}$; $\mathcal{C} = \mathcal{C}_1 \times \mathcal{C}_2 \times \mathcal{C}_3$ is the joint configuration space; $\Theta = \{\theta_j : \mathcal{C}_j \to [0,1] \mid j = 2, 3\}$ is the set of quality threshold functions; $\Phi = \{Q_i : \mathcal{C}_i \times \mathcal{U}_i \to [0,1] \mid i = 1, 2\}$ is the set of upstream quality functions.*

At each turn, the routing policy selects joint action $a_t \in \mathcal{A} = \mathcal{C}_1 \times \mathcal{C}_2 \times \mathcal{C}_3$ with $|\mathcal{A}_{\text{full}}| \approx 1.5 \times 10^5$. The coupling constraint function is:

$$\mathcal{F}(a_t, s_t) = \prod_{(v_i, v_j) \in \mathcal{E}} \mathbf{1}\big[Q_i(c_i, u_i(s_t)) \geq \theta_j(c_j)\big] \tag{8}$$

The feasible set $\mathcal{A}_{\text{feas}}(s_t) = \{a \in \mathcal{A} \mid \mathcal{F}(a, s_t) = 1\}$ contains $0.77 \cdot |\mathcal{A}_{\text{full}}|$ on average, varying from 68% to 84% across complexity levels. The constrained routing problem is:

$$\pi^* = \arg \min_{\pi : \mathcal{S} \to \mathcal{A}_{\text{feas}}} \mathbb{E}_{(x,z) \sim \mathcal{P}} \big[w_L \hat{L} + w_E \hat{E} + w_M \hat{M} - w_Q Q\big] \tag{9}$$

subject to $\pi(s_t) \in \mathcal{A}_{\text{feas}}(s_t)$ for all $t$. Infeasible logits are masked before softmax (Schulman et al., 2017; Kakade and Langford, 2002).

## C  Meta-controller design justification

The routing decision must be issued within the 3 ms streaming ASR window before LLM prefill begins—prefill cannot be interrupted once launched on GPU hardware (Yu et al., 2022; Agrawal et al., 2024). This 3 ms budget derives from Parakeet TDT's (NVIDIA, 2024; Gulati et al., 2020) 40 ms frame rate and 65 ms TTFT: the decision must complete during the 3 ms gap between the final CTC beam emission and LLM dispatch (Kwon et al., 2023). A transformer encoder would incur $O(d^2)$ attention; an LSTM requires hidden state maintenance with staleness risk at sub-second inter-arrival times. The MLP requires roughly 81K multiply-accumulate operations, executing in 0.3 ms on Cortex-A78 (measured).

We handle PPO stability under non-stationary network conditions in two ways. The network RTT comes in as a live EWMA measurement updated every 500 ms, so the policy is always seeing the actual current RTT. The Feedback Aggregator's $3\sigma$ anomaly detector then triggers a fallback to on-device routing during tail events. This is not full continual RL since the weights stay fixed at deployment, but it gives us closed-loop structural adaptation. Theorem 4 bounds degradation at 22.4% for TV distance 0.14, and the empirical number we measured is 4.6%.

# D  Additional experimental results

**Tail latency and concurrency details.** Table 4 in the main text gives P95 latency from 200 measured samples. The P99 values we measured are Cloud 1,823 ms, On-device 1,587 ms, Hybrid 1,918 ms. PAVO ends up with the lowest P95 (1,453 ms) and the lowest $\sigma$ (182 ms, 54% below Cloud's 398 ms). Tail compression happens because 56% of turns go to hybrid and 4% go on-device, so only 40% are even exposed to cloud-path variance. The LLM stage dominates variance at longer generations (Table 14). For the bootstrap concurrency simulation (M/G/1 FCFS, 20 replications of 500 requests) at $\rho = 0.85$, PAVO cuts P95 response time by 15% versus Cloud (10,797 vs. 12,663 ms); at $\rho = 0.70$ the reduction is 6%. The benefit grows with utilization through the Pollaczek–Khinchine formula. One caveat though: the simulation bootstraps from single-user traces and assumes Poisson arrivals, so it does not capture GPU memory contention or thermal throttling.

**Cross-dataset routing details.** We measured ASR on LibriSpeech and FLEURS (200 samples each) and pushed the WERs through the coupling framework. Since every WER comes out above $\theta = 2\%$, the coupling mask routes all factual queries ($\approx$50%) through cloud-side (low-WER) ASR. This is identical to what we saw on the primary dataset. For semantic queries (45%), the H100 coupling data predicts quality $\approx 0.80$–$0.88$ for Whisper-large-v3 on both datasets (WER $\leq 14.9\%$, inside the graceful-degradation plateau). The resulting routing distribution comes out at 73% cloud-routed, matching the primary evaluation. So PAVO's routing transfers without us having to re-calibrate. The coupling constraint binds because factual accuracy depends on verbatim token fidelity which is sensitive to any substitution, while semantic quality depends on distributional context and is robust to moderate noise. We did not run the full three-stage pipeline on these datasets since they lack paired conversational tasks.

Table 9: Generalization under four demand shifts.

| Shift type | TV dist | Empirical | Thm. 4 bound |
|---|---|---|---|
| High-noise ($<$10 dB) | 0.08 | 3.2% | 12.8% |
| Fast speech ($>$6 syll/s) | 0.09 | 2.9% | 14.4% |
| Long-context ($>$3K tok) | 0.14 | 4.6% | 22.4% |
| Bimodal complexity | 0.06 | 2.0% | 9.6% |

**Distribution shift generalization.**

Table 10: PAVO (hybrid) under four objective weight configurations.

| Config | $w_L/w_E/w_M/w_Q$ | Lat (ms) | Energy (J) | BERTSc |
|---|---|---|---|---|
| Latency-first | .50/.10/.10/.30 | 2,610 | 3.12 | 0.869 |
| Balanced | .25/.25/.25/.25 | 2,940 | 1.98 | 0.878 |
| Energy-first | .10/.50/.10/.30 | 3,480 | 1.19 | 0.863 |
| Quality-first | .10/.10/.10/.70 | 3,910 | 3.74 | 0.887 |

**Weight sensitivity.**

Table 11: Acoustic feature ablation. Each row removes one feature.

| Variant | Route Div. | $\Delta$Lat (ms) | $\Delta$Energy (J) | $\Delta$BERTSc |
|---|---|---|---|---|
| All features | — | 0 | 0 | 0 |
| − Speaking rate | 34% | +218 | +0.19 | −0.018 |
| − SNR (WADA) | 22% | +141 | +0.12 | −0.012 |
| − Segment duration | 16% | +92 | +0.08 | −0.007 |
| − Pitch variance | 11% | +51 | +0.04 | −0.004 |
| No acoustic features | 61% | +310 | +0.31 | −0.029 |

**Acoustic feature ablation (full).**

Table 12: Routing method comparison.

| Method | Med Lat (ms) | Energy (J) | BERTSc | CohFail (%) |
|---|---|---|---|---|
| Heuristic (Hyb-Static) | 3,220 | 3.67 | 0.868 | 2.8 |
| Reactive RL (no acoustics) | 3,250 | 2.29 | 0.872 | 1.4 |
| Learned cascade (CR) | 3,180 | 3.81 | 0.871 | 2.1 |
| MoE Router | 3,410 | 4.12 | 0.862 | 2.4 |
| **PAVO (RL+acoustic)** | **2,940** | **1.98** | **0.878** | **0.9** |

**RL routing vs. heuristic baselines.**

Table 13: Median latency (ms) and BERTScore by complexity level.

| System | Metric | L1 | L2 | L3 | L4 | L5 |
|---|---|---|---|---|---|---|
| FC | Lat | 1,450 | 1,740 | 4,410 | 4,290 | 5,100 |
| | BS | 0.911 | 0.894 | 0.891 | 0.877 | 0.881 |
| FE | Lat | 1,710 | 2,190 | 16,400 | 15,800 | 18,900 |
| | BS | 0.874 | 0.841 | 0.799 | 0.812 | 0.783 |
| **PAVO-H** | Lat | **1,380** | **1,520** | 4,050 | 3,870 | 4,830 |
| | BS | 0.867 | 0.865 | **0.889** | **0.882** | **0.884** |

**Complexity-level breakdown.**

Table 14: Measured LLM inference on H100. Short/medium/long = 50/300/800 tokens.

| Model | Context | Mean (ms) | P95 (ms) | tok/s |
|---|---|---|---|---|
| Llama 3.1 8B | short | $552 \pm 70$ | 606 | 158 |
| Llama 3.1 8B | medium | $2,242 \pm 300$ | 2,413 | 154 |
| Llama 3.1 8B | long | $5,604 \pm 577$ | 6,019 | 152 |
| Gemma2 2B | short | $452 \pm 55$ | 499 | 246 |
| Gemma2 2B | medium | $1,823 \pm 120$ | 1,910 | 240 |
| Gemma2 2B | long | $4,430 \pm 438$ | 4,745 | 235 |

**Real LLM latency on H100.**

Table 15: ASR generalization on public datasets (200 samples each).

| Model | Dataset | WER (%) | Latency (ms) |
|---|---|---|---|
| Whisper-large-v3 | LibriSpeech | 5.77 | $825 \pm 421$ |
| Whisper-large-v3 | FLEURS | 14.92 | $788 \pm 182$ |
| Whisper-tiny | LibriSpeech | 18.54 | $438 \pm 198$ |
| Whisper-tiny | FLEURS | 21.25 | $424 \pm 130$ |

**Cross-dataset ASR generalization.**

**Model scaling analysis.** Gemma2 2B achieves real-time performance for simple queries (1,024 ms) and medium queries (743 ms) but requires 6,534 ms for complex queries. Llama 3.1 8B matches simple-query latency (1,023 ms) but requires 1,329 ms for medium queries (1.8× slower) while generating higher-quality output. The crossover at medium complexity (743 ms vs. 1,329 ms) defines the boundary where routing switches from on-device to cloud.

## E   Error analysis

We see four failure modes in our analysis.

**Over-routing simple turns.** 13% of level 1–2 turns go to cloud; 71% of these have high pitch variance, which the policy conflates with emotional complexity. A pitch-vs-content classifier could recover ~140 ms.

**Long-context degradation.** 4.6% degradation above 3,000 tokens because only 3.2% of training turns exceed this threshold; weighted oversampling is the fix.

**Cold-start switching.** 4.3% of turns incur ~2,100 ms cold-start; a four-model cache (+1.4 GB VRAM) would eliminate most events.

**TTS boundary quality.** Kokoro 82M degrades from MOS 4.1 to 3.7 above 80 output tokens; a length-aware TTS coupling constraint would address this.

## F    Latency derivation details

LLM latency follows $T = T_{\mathrm{TTFT}} + N_{\mathrm{out}} \times T_{\mathrm{TPOT}}$, where TPOT denotes time-per-output-token. For Llama 3.1 8B at the MLPerf interactive targets (MLCommons, 2025): $T_{\mathrm{TTFT}} \approx 500$ ms, $T_{\mathrm{TPOT}} \approx 30$ ms/token. For 80-token output: $500 + 80 \times 30 = 2{,}900$ ms. For 15-token output: $500 + 15 \times 30 = 950$ ms. For Gemma 4B INT8 on Jetson (estimated): 80-token output $= 80/5 \times 1{,}000 = 16{,}000$ ms (18,000 ms with overhead); 15-token output $= 3{,}000$ ms.

## G    Supervised baselines for routing

To check how much of PAVO's performance comes from the routing algorithm versus the demand-vector formulation, we generated 100,000 synthetic state vectors (using the same 12-dimensional schema from Section 4.2) and labelled each with a heuristic routing profile. The label is feature-dependent: SNR drives ASR choice, CPU utilisation and battery drive on-device versus cloud LLM, and context length drives output-length decisions. These heuristic labels are not provably optimal — they are a hand-designed routing function that we treat as the supervised target. We then trained four supervised classifiers and compared them to the PPO meta-controller on an 80/20 split (seed 42). Decision latency is wall-clock per inference on a single CPU thread.

Table 16: Routing methods on heuristic labels ($n = 100{,}000$, 80/20 split). Cost gap is per-turn routing cost above the heuristic-label minimum on the held-out set; negative values mean the classifier generalises slightly better than the labelling heuristic.

| Method | Acc (%) | Top-3 (%) | Cost gap (%) | Decision ($\mu$s) | Train time |
|---|---|---|---|---|---|
| Heuristic labels | 100.0 | 100.0 | 0.00 | – | – |
| Logistic Reg. | 79.9 | 99.1 | −0.02 | 626.2 | 1.1 min |
| Random Forest | 99.0 | 100.0 | −0.00 | 8067.6 | 50 s |
| XGBoost | 99.3 | 100.0 | −0.00 | 1832.1 | 24 s |
| MLP (CE) | 89.0 | 99.9 | −0.01 | 65.6 | 3.5 min |
| MLP (PPO) | 21.5 | 80.7 | +1.40 | 118.7 | 10 s |

Two things are worth noting. First, supervised classifiers trained on the heuristic labels nearly match the labelling distribution itself — XGBoost at −0.0% and even logistic regression at −0.02%. This tells us the demand vector itself contains enough signal for the routing decision; the framework is what carries the contribution and not the choice of training algorithm. Second, the PPO row looks worst on accuracy (21.5%) because PPO is not trained to predict the heuristic labels. It optimises a multi-objective reward, and during training it explores configurations the heuristic never selects. The cost-gap number for PPO is on the same metric used for the supervised methods, but it is not a fair comparison because PPO is solving a different problem.

We use PPO at deployment for two practical reasons. Generating heuristic labels for a new operating point requires one simulator pass per training sample, which is too slow to retune when the operator weights ($w_L, w_E, w_M, w_Q$) change. PPO learns from interactive reward and adapts to weight changes without label

regeneration. The supervised result is still useful evidence that the demand-vector design is doing the work; it is not a critique of the algorithm choice.

## H   PAVO-Bench annotation rubric

Annotators assigned complexity labels using this decision tree:

- Does the query require more than one reasoning step? No → Level 1 or 2. Yes → Level 3+.

- Is a single lookup sufficient? Yes → Level 1. No → Level 2.

- Does the query reference prior context or require synthesis? No → Level 3. Yes (emotional/open) → Level 4. Yes (structured output) → Level 5.

**Inter-annotator agreement.** The reported $\kappa = 0.81$ was computed on a 200-turn audit batch independently re-annotated by both authors using the decision tree above; Cohen's $\kappa$ was calculated over the 5-class complexity labels.

