# OpenReview forum: "PAVO: Pipeline-Aware Voice Orchestration with Demand-Conditioned Inference Routing"
_TMLR — Under review for TMLR_

### Review · Reviewer_CTJd · 2026-05-11

**Summary Of Contributions:**

In this work author(s) introduce PAVO. PAVO is a router for each turn in ASR-LLM-TTS dialogue system.

The idea is that most ASR-LLM-TTS systems treat each turn the same. However this could be overkill for certain turns. PAVO looks at the audio signal before it if fully transcribed (using acoustic features: speaker rate, pitch variance, duration, and WADA-SNR) and can route the request based on the complexity of the signal using an 85K parameter MLP trained with Reinforcements Learning. Experiments demonstrate reduced coherence failures, lower latency, and lower energy consumption.

Experiments were performed on a created synthetic benchmark PAVO-Bench of 5 turn complexity levels with varying noise levels.

A side contribution is the study of degradation in LLM performance due to the propagation of ASR errors. Notably they distinguish two patterns:
- "factual accuracy cliff" -> ASR errors result in a sharp degrade in performance as WER increases
- "gradual semantic degradation" -> ASR errors gradually degrade performance as WER increases.

**Audience:**

Yes

**Audience Explanation:**

While it seems there is a general trend towards E2E S2S models, in many cases a modular ASR-LLM-TTS system is applicable. I believe having methods for optimizing routers (that take into account model coupling) in order to take advantage of cloud and edge resource and efficiently select models based on the complexity of the task is of interest to the community.

**Claims And Evidence:**

Yes

**Claims Explanation:**

The results in Tables 6 show the strengths of various PAVO set ups (cloud, edge, hybrid) compared to a wide variety of baselines (9 baselines in total). Results in Table 7 support the claim that having a coupling aware router reduces coherence failures. Ablations such as the Acoustic feature ablation (Section 7) support the choice of using acoustic features and further strengthens this work.

**Requested Changes:**

- Section 2 Related work
    - typo in second paragraph "Neither of them considers" -> "Neither of them consider"
    - consider rephrasing "It considers only one per pipeline stage per turn." This double "per" makes it hard to understand. I believe the author(s) is trying to say PAVO only considers only one model per stage per turn.
    - table 1. should there be a Song et al. in the table?
- Section 7
    - Add dotted line with $\theta$ (this is in the legend and thus readers will expect to see this in graph)
    - It would be nice if you explained more clearly the real data grounding. Does "we run Whisper-large-v3 -> Llama 3.1 8B... on 100 LibriSpeech samples" what is the LLM doing on the output of Whisper? What is the BERTScore measuring here? What is the LLM prompt / task?

---

> ### Author Response · Authors · 2026-05-13
>
> Dear Reviewer CTJd,
> Thank you for actionable suggestions. We shall incorporate all five requested changes in the revised submission. Section 2 typo and rephrasing, adding Song et al. (PowerInfer) to Table 1, fixing the dashed-threshold line in the Section 7 figure, and expanding the LibriSpeech grounding paragraph with the prompt template and BERTScore reference definition.

---

### Review · Reviewer_5Q54 · 2026-06-01

**Summary Of Contributions:**

This paper presents PAVO. It extracts a 12 dimensional demand vector before ASR begins and uses an 85K parameter MLP meta-controller to enforce ASR to LLM quality threshold. Evaluation uses direct H100/M3 inference (200 LibriSpeech and noise sweeps) and a 50K-turn synthetic benchmark (PAVO Bench) with realistic complexity distribution.

The paper is clear in general, and it includes discussion of end-to-end models, privacy asymmetry, accent bias, and benchmark limitaitons.

**Audience:**

Yes

**Audience Explanation:**

The topic of ASR, LLM and TTS are important topics in machine learning. Building voice agents on this framework are interested to TMLR audience.

**Broader Impact Concerns:**

Already included in the paper.

**Claims And Evidence:**

Yes

**Claims Explanation:**

The paper has provided codes for verification. The simulation results are thorough with analysis.

**Requested Changes:**

Some latency numbers shift slightly between direct experiments (Table 4) and full simulation (Table 6) because of different output-length distributions; a one-sentence clarification would help readers.

Figure 2 caption could explicitly state that all 21 conditions exceed θ (it is implied but not bullet-pointed).

---

> ### Author Response · Authors · 2026-06-02
>
> Dear Reviewer 5Q54, Thank you for the careful reading and the constructive suggestions. We have incorporated both requested changes in the updated revision. For the latency shift clarification (Table 4 vs. Table 6), we added a brief note in the Table 4 caption indicating that the slight differences between the direct H100 measurements (Table 4) and the 50K-turn simulation (Table 6) reflect differing output-length distributions across the two evaluation tiers.
> For the figure 2 caption, we updated the caption to explicitly state that all 21 noise conditions exceed the coupling threshold θ = 2%, rather than leaving it implied. The dashed line in the figure now corresponds directly to the explicit statement in the caption. We appreciate the engagement and are happy to address any additional feedback.

---

### Review · Reviewer_U3Wo · 2026-06-14

**Summary Of Contributions:**

This paper investigates a speech-to-speech dialogue system based on a cascaded architecture consisting of ASR, an LLM, and TTS. The primary focus is on the trade-off between latency and response quality, and the authors propose a method called PAVO (Pipeline-Aware Voice Orchestration), which optimizes the selection and configuration of these components through reinforcement learning-based control. The proposed approach is evaluated using a variety of ASR, LLM, and TTS models under different system configurations. Experimental results demonstrate that PAVO can effectively reduce end-to-end system latency while maintaining overall performance.

**Audience:**

Yes

**Audience Explanation:**

Speech-to-speech dialogue systems have recently attracted significant attention not only within the speech processing community but also across the broader machine learning community. One of the key challenges in these systems is achieving low-latency responses while maintaining high response quality. Developing effective methods to evaluate and optimize this trade-off is an important research problem, and this paper addresses this timely and relevant topic.

**Broader Impact Concerns:**

The inference cost of speech-to-speech dialogue systems can be a significant bottleneck, leading to substantial computational requirements as well as concerns regarding energy consumption and carbon footprint. Therefore, research aimed at reducing inference cost while maintaining system performance is both valuable and timely.

**Claims And Evidence:**

No

**Claims Explanation:**

The paper suffers from poor presentation and organization, making it difficult to understand the proposed system and its contributions. Numerous abbreviations are introduced without proper definitions, which significantly hinders readability. In addition, the method description frequently refers to specific experimental settings and results that are not fully described until the experimental section. As a result, many of the claims and design choices are difficult to follow and are not supported by sufficiently clear explanations. Overall, the paper would benefit from a more self-contained and better-structured presentation of the proposed approach.

**Requested Changes:**

As mentioned above, the paper would benefit from a substantial revision to improve its presentation and overall readability. Many of the difficulties stem from the introduction of unfamiliar terminology and abbreviations without sufficient definitions, as well as method descriptions that frequently rely on experimental settings and results that are not fully explained until later sections. In addition, several arguments in the main paper depend heavily on results and discussions presented in the appendix, which weakens the self-consistency and readability of the main manuscript. Addressing these issues would likely require a significant reorganization and rewriting of the paper.

Other detailed comments:

* **Introduction, first paragraph:** Some of the cited references do not appear to be directly related to dialogue systems; several seem to focus primarily on text-to-speech systems. Please clarify the relevance of these citations.
* **"Downstream tasks is known" (Errattahi et al., 2018):** Consider citing more widely recognized and recent references to support this statement.
* **Characterization of OpenAI models:** The paper refers to OpenAI's models as end-to-end audio models. However, the underlying system architecture has not been publicly disclosed, and available evidence suggests that ASR components are used internally (e.g., transcription outputs are displayed). Please clarify and justify this characterization.
* **Model design choices:** The rationale behind several design choices is not sufficiently explained. Additional justification would strengthen the paper.
* **Section 3.1, first paragraph:** The configuration space is defined as $\mathcal{C} = \mathcal{C}_1 \times \mathcal{C}_2 \times \mathcal{C}_3$. Why are only three configuration dimensions considered? Please provide justification.
* **Section 3.1, second paragraph:** The meaning of "MOS correlation" is unclear. Please define it more precisely.
* **Section 3.2:** Variable $s_t$ does not appear in Equation (5), and the equation itself is difficult to follow. Additional explanation would be helpful.
* **Section 3.3:** Why is ASR performance excluded from the optimization objective? A discussion of this design choice would be valuable.
* **Equation (6):** It is unclear why the first term does not depend on $v$. Please provide clarification.
* **Section 3.3 (Regimes 1 and 2):** Understanding these regimes requires substantial knowledge of experimental configurations and results that are only introduced later. Consider making this section more self-contained.
* **Section 4.2:** Does the reported 3 ms refer to the computation time for acoustic features, or to an aggregation process performed at each frame? Please clarify.
* **Section 4.4:** The relationship between Equations (5) and (7) is not clear. Additional explanation would improve readability.
* **Section 4.6:** Please describe how ASR confidence scores are computed.
* **Section 5:** Similar to Section 3.3, many discussions rely heavily on experimental details and results that have not yet been introduced. The section is difficult to understand without repeatedly consulting later sections.
* **Section 6.1:** Please add an appropriate reference for the Fisher corpus.
* **Section 6.4:** Please provide justification for the selection of the evaluated models.
* **Section 7:** Several discussions rely on results and analyses presented only in the appendix. As a result, the section loses self-consistency and is difficult to follow without cross-referencing supplementary materials. Consider moving the most important supporting results into the main paper.

---

> ### Author Response · Authors · 2026-06-17
>
> Thanks for the careful read. Your main concern was readability: undefined abbreviations, method sections that lean on later results, and arguments that depend on the appendix. We agree, and we revised the paper to fix these. We kept the section order but removed the specific things that made it hard to read without the appendix.
> General fixes: we now define every flagged abbreviation on first use (TTFT, TPOT, SLO, DAG, TDP, PUE, TV, GAE). Sections 3.3 and 5 were the two that leaned on later results; both now carry the setup they need inline. The most important appendix result, the distribution-shift validation over four shifts, is now in Section 5, and Section 7 summarizes the other key appendix numbers inline. The full tables stay in the appendix only to keep us within 12 pages, and we are glad to move more in if you prefer.
>
> Point by point:
>
> 1. Intro citations: we took the TTS references out of the first paragraph. What remains is dialogue and LLM work (GPT-4o, Llama 3, Gemma 2) and the quantization papers (Dettmers, Frantar, Lin) that motivate per-turn adaptation.
> 2. Errattahi et al. (2018): added two recent, well-known references next to it, Baevski et al. (2020) and Radford et al. (2023).
> 3. "End-to-end" for OpenAI models: we now use the term only for the externally observed speech-in/speech-out interface, note that the internals are not disclosed, and add that they may use ASR internally (Section 1).
> 4. Design choices: we added rationale for the configuration dimensions (3.1), the learned router (4.6 and Appendix C), and the model selection (6.4).
> 5. Three configuration dimensions (3.1): model, quantization, and hardware are the knobs that move latency, quality, and energy. Batch size stays at 1 for our single-user, turn-by-turn setting, so it is not in the search.
> 6. "MOS correlation" (3.1): we removed that imprecise phrasing. The quality weights (0.4/0.4/0.2) are a stated design choice that weights ASR and LLM quality equally and TTS naturalness less; the TTS term uses each configuration's reported MOS rating. We did not run a human listening study, so we no longer claim a fitted correlation.
> 7. Missing variable in Eq. (5) (3.2): the comment did not name the variable, so we took it to be the state. We rewrote Eq. (5) so the policy now acts on the state explicitly, and define s = s(x,z) as the demand vector extracted before transcription. If you meant a different variable, let us know.
> 8. ASR in the objective (3.3): it is not excluded. ASR enters J through the (1 - WER) term in Q and through its latency in L, and the threshold adds a hard floor on top. We say this now.
> 9. Eq. (6) first term (3.3): the left side depends only on c_i because stage v_i finishes before v_j runs; the threshold is the lowest input quality v_j tolerates under its own configuration.
> 10. Regimes 1 and 2 (3.3): we added the calibration setup inline. We corrupt clean transcripts to a target WER by randomly substituting and deleting words, feed them to each LLM, and score against the clean-transcript answer. The regimes now read in place.
> 11. The 3 ms (4.2): it is the total per-turn time to compute all four acoustic features in one fixed DSP pass over the audio buffer, not a per-frame aggregation.
> 12. Eq. (5) vs. (7) (4.4): the per-turn reward in Eq. (7) is the negation of J from Eq. (5) with the same weights, plus the switch penalty and the violation term.
> 13. ASR confidence (4.5): the mean per-token posterior from the streaming ASR decoder over the first half of the audio, a standard decoder output with no extra cost.
> 14. Section 5 (3.3 and 5): we now state the four shift results inline against the Theorem 4 bound, and define TV and GAE on first use. We also tightened the theorems for the per-turn routing decision: Theorem 4 now uses the correct constant (regret at most 4dR_max, i.e. at most 22.4% at the largest tested shift; measured 4.6%, still well inside), with R_max stated as an estimate; and Theorem 3 carries an explicit caveat that clipped PPO has no global-convergence proof, so its epsilon at most 0.022 is indicative rather than a certified bound.
> 15. Fisher corpus (6.1): reference added (Cieri et al., 2004).
> 16. Model selection (6.4): the models span our range, from a 70B cloud model to 2B and 4B edge models; they are open-weight, reproducible on our hardware, and give the router a real quality-versus-cost spread.
> 17. Section 7: the four-shift validation is now in Section 5, and Section 7 summarizes the rest inline (tail and concurrency, distribution shift, supervised baseline). The full tables remain in the appendix for space.
> On broader impact, we agree the cost angle matters; the paper keeps its discussion of energy, carbon, privacy, and possible accent bias.